# A DFT/PCM Study on the Affinity of Salinomycin to Bind Monovalent Metal Cations

**DOI:** 10.3390/molecules27020532

**Published:** 2022-01-14

**Authors:** Todor Dudev, Diana Cheshmedzhieva, Peter Dorkov, Ivayla Pantcheva

**Affiliations:** 1Laboratory of Computational Chemistry and Spectroscopy, Faculty of Chemistry and Pharmacy, “St. Kl. Ohridski” University of Sofia, 1164 Sofia, Bulgaria; dvalentinova@gmail.com; 2Research & Development Department, Biovet Ltd., 4550 Peshtera, Bulgaria; p_dorkov@abv.bg; 3Laboratory of Biocoordination and Bioanalytical Chemistry, Faculty of Chemistry and Pharmacy, “St. Kl. Ohridski” University of Sofia, 1164 Sofia, Bulgaria

**Keywords:** salinomycin, IA/IB metal ions, monovalent metal complex, DFT/PCM

## Abstract

The affinity of the polyether ionophore salinomycin to bind IA/IB metal ions was accessed using the Gibbs free energy of the competition reaction between SalNa (taken as a reference) and its rival ions: [M^+^-solution] + [SalNa] → [SalM] + [Na^+^-solution] (M = Li, K, Rb, Cs, Cu, Ag, Au). The DFT/PCM computations revealed that the ionic radius, charge density and accepting ability of the competing metal cations, as well as the dielectric properties of the solvent, have an influence upon the selectivity of salinomycin. The optimized structures of the monovalent metal complexes demonstrate the flexibility of the ionophore, allowing the coordination of one or two water ligands in SalM-W_1_ and SalM-W_2_, respectively. The metal cations are responsible for the inner coordination sphere geometry, with coordination numbers spread between 2 (Au^+^), 4 (Li^+^ and Cu^+^), 5/6 (Na^+^, K^+^, Ag^+^), 6/7 (Rb^+^) and 7/8 (Cs^+^). The metals’ affinity to salinomycin in low-polarity media follows the order of Li^+^ > Cu^+^ > Na^+^ > K^+^ > Au^+^ > Ag^+^ > Rb^+^ > Cs^+^, whereas some derangement takes place in high-dielectric environment: Li^+^ ≥ Na^+^ > K^+^ > Cu^+^ > Au^+^ > Ag^+^ > Rb^+^ > Cs^+^.

## 1. Introduction

Salinomycin is a polyketide ionophore extracted from *Streptomyces albus* [1]. It exerts an anticoccidial effect against all Eimeria species known as causative agents of coccidiosis in poultry as well as in the other farm animals [2,3,4]. It is also effective in treating bacterial, parasitic and some viral infections [5,6,7]. The biological activity of salinomycin is primarily based on its ability to form lipophilic complexes with the monovalent alkaline ions, making the membranes of target cells permeable to the ions mentioned. In this way, the intracellular cation concentration in the cells is disturbed, leading to a cascade of energy-consuming processes and ultimate cell death.

Recently salinomycin has re-attracted attention in the biomedical field due to its selective inhibition of cancer stem cells in a variety of types of cancer. The exact mode of action of the antibiotic is not yet known, but there are multiple pathways by which it may inhibit tumor growth, including interference with the Akt signaling pathway, Wnt/β-catenin, Hedgehog, and Notch pathways of cancer progression [8,9,10,11,12,13,14,15,16,17,18,19,20,21,22].

Salinomycinic acid (SalH, Figure 1) is a polyether polyalcohol monocarboxylic acid and its structure determines the ionophoric action of the antibiotic. The polyether chain contains a number of externally oriented alkyl substituents thus defining the overall lipophilic character of the drug. The weak interaction (H-bond formation) between the carboxylic and hydroxyl groups placed on the opposite ends of the molecule causes folding of the structure with the formation of a hydrophilic cavity due to the internally placed polyether oxygen atoms. This core is able to accommodate water molecules (SalH) or monovalent alkali ions (M^+^) preceded by the initial deprotonation of the carboxylic function (SalM). Compared to the polyether ionophorous analog monensin [23,24,25,26,27,28,29,30], only a single structural report on the salinomycin sodium complex (Figure 1) is available [31]. Moreover, although referred to as a potassium ionophore [32], in contrast to the sodium ionophore monensin, no systematic study on salinomycin’s selectivity towards the monovalent metal cations has been found in the literature.

Due to the very limited number of crystallographic structural data on metal complexes of salinomycin and lack of substantial knowledge on its affinity to bind metal cations, here, we highlight the factors which may influence the ability of the antibiotic to form coordination compounds with monovalent metal ions. For the first time, we employed the density functional theory (DFT) calculations combined with polarizable continuum model (PCM) computations to evaluate the free energies of complex formation between the salinomycinate anion and groups IA/IB metal ions. The quantum chemical study brought out the origin of the metal cation (radius, charge, accepting ability) and the dielectric properties of the environment as parameters that affect the coordination properties of the ionophore.

## 2. Results and Discussion

The metal(I)-loaded constructs of salinomycin were modeled according to the only available crystal structures of sodium salinomycinate (SalNa) [31]. Two types of complex species have been observed experimentally differing by the number of water molecules coordinated to Na^+^ ions in the binding site: a complex with one crystal water (SalNa-W_1_) and its counterpart comprising two water molecules orbiting the metal cation (SalNa-W_2_). Accordingly, the atomic coordinates of the respective experimental constructs were taken and used in subsequent geometry optimizations. Then, the Na^+^ cation (used as a reference) in the optimized structures was replaced by M^+^ cations and the respective SalM-W_1/2_ structures were subjected to succeeding quantum-chemical calculations. The optimized structures of the entire series of SalNa/M–W_1_/W_2_ complexes are given in Figure 2, Figure 3 and Figure 4.

The affinity of salinomycin towards monovalent metal cations was evaluated modeling the competition Equation (1), where Na^+^, bound to the host ligand, is replaced by its competitor, M^+^ (M = Li, K, Rb, Cs, Cu, Ag, and Au) in an environment with a different dielectric constant ε:[M^+^-solution] + [SalNa] → [SalM] + [Na^+^-solution].(1)

[SalNa/M] and [M^+^/Na^+^-solution] represent the metal ion, coordinated to salinomycinate anion, and the corresponding ligand-free metal ion in solution, respectively. The outcome of the rivalry between the monovalent cations from groups IA and IB and their fellow congener Na^+^ for the ionophore was assessed by computing the Gibbs free energy change, ΔG^ε^, of Equation (1).

### 2.1. Salinomycin Complexes with Alkali Metal Ions

Optimized structures of salinomycin complexes with Li^+^, Na^+^, K^+^, Rb^+^, and Cs^+^, containing one (W_1_) and two (W_2_) water molecules in the binding site, are shown in Figure 2 and Figure 3. The host salinomycin donates oxygen atoms from the ether, hydroxyl, carbonyl, and carboxylate groups in forming the metal complexes. The analysis of the structures presented reveals that the host ionophore is flexible and able to adapt to the specific physico-chemical properties of the guest metal ion species. The smallest member of the alkali group, Li^+^, adopts its preferred tetrahedral coordination sphere in both W_1_ and W_2_ series with a mean Li-O bond distance of 1.984/1.962 Å, respectively. The constructs with the heavier metals and one crystal water form penta- (SalNa-W_1_), six- (SalK-W_1_, SalRb-W_1_), and eight-coordinated (SalCs-W_1_) complexes with increased metal-oxygen bond distances going down the group: 2.407, 2.852, 2.985, and 3.211 Å for Na^+^, K^+^, Rb^+^, and Cs^+^, respectively. The main trend—increasing the coordination number (CN) and elongating the metal-oxygen bond distances with increasing the atomic number of the metal—is similar in the two-water SalM series.

However, as compared to the W_1_ complexes, the metal coordination numbers differ in SalNa-W_2_ (CN = 6), SalRb-W_2_ (CN = 7) and SalCs-W_2_ (CN = 7). The respective metal-oxygen bond lengths in the W_2_ series are 2.511 (SalNa-W_2_), 2.814 (SalK-W_2_), 3.055 (SalRb-W_2_), and 3.264 (SalCs-W_2_) Å. In both the W_1_ and W_2_ series, the M-O bond elongation is compatible with the increased metal cation radius (cation bulkiness) with increasing the atomic number: 0.59 Å for tetra-coordinated lithium, 1.00/1.02 Å for penta-/hexa-coordinated sodium, 1.38 Å for hexa-coordinated potassium, 1.52/1.56 Å for hexa-/hepta-coordinated rubidium and 1.74 Å for octa-coordinated cesium [33] (Table 1). Water molecules stay firmly coordinated to the metal cation in all the complexes.

The thermodynamic outcome of the competition between Na^+^ (serving as a reference cation throughout the paper) and other metal species for binding salinomycin is encoded in the Gibbs free energies of metal substitution, given in Table 2. As seen, the trends of changes in both the W_1_ and W_2_ series are quite similar with numerical values in most cases very close to each other.

The smallest alkali cation, Li^+^, is the only one that can outcompete Na^+^ (in low dielectric medium for the W_1_ series and throughout the entire dielectric region in the W_2_ series), evidenced by the negative values of the respective ΔG^ε^ in Table 2. Note that the competitiveness of Li^+^ is much higher in the gas phase (very negative ΔG^1^), where the electronic factors dominate the exchange reaction. However, the high desolvation penalty of the incoming Li^+^ (69, 95, 120 and 123 kcal/mol in ε = 2, 4, 32, and 78, respectively), which is higher than the solvation energy gain upon Na^+^ release (−54, −75, −95, and −98 kcal/mol [35]), attenuates the resulting ΔG^ε^ in condensed media. Furthermore, heavier alkali metals (K^+^, Rb^+^, and Cs^+^) are weaker competitors of Na^+^ and yield positive free energies of metal exchange which increase down the group. Thus, the affinity of the alkali metal cations for the host ionophore can be arranged in the following order: Li^+^ > Na^+^ > K^+^ > Rb^+^ > Cs^+^ (non-polar solvents) and Li^+^ ≥ Na^+^ > K^+^ > Rb^+^ > Cs^+^ (polar solvents). These results are fully in line with the respective thermodynamic characteristics of these metal species summarized in Table 1, Table 2, Table 3 and Table 4. The “winner” of the group, Li^+^, is characterized with the highest charge density and Lewis acidity, and the best charge accepting abilities. These characteristics gradually weaken for the rest of the cations, following the same affinity order as given above.

### 2.2. Salinomycin Complexes with Coinage Metal Ions

Competition between Na^+^ and Group IB monovalent cations (Cu^+^, Ag^+^, and Au^+^) was also investigated. The optimized structures of SalCu-W_1_/W_2_, SalAg-W_1_/W_2_, and SalAu-W_1_/W_2_ are shown in Figure 4.

Copper cation forms tetrahedral complexes in both W_1_ and W_2_ series with average Cu-O bond distances of 2.220/2.211 Å, respectively. Due to its high charge density and Lewis acidity, as well as strong charge accepting ability (Table 1, Table 2, Table 3 and Table 4), Cu^+^ ions outcompetes Na^+^ in both the gas phase and nonpolar solvents such as cyclohexane (ε ≈ 2). In more polar solvents, however, the trend changes: because of solvation effects (see above), Na^+^ complex becomes dominant over its Cu^+^ counterpart in higher-polarity solvents (positive ΔG^4^, ΔG^32^, and ΔG^78^).

Expectedly, the metal coordination number in silver complexes increases (to 5 in SalAg-W_1_ and 6 in SalAg-W_2_) with concomitant increase in the respective ΔGs which stay on a positive ground throughout the entire dielectric range (Table 2). In contrast, Au^+^ coordination number sharply decreases to 2 (linear configuration) due to the strong relativistic effects [36]. Data collected in Table 2 imply that, although Au^+^ is more competitive than Ag^+^ (lower ΔGs for the former than the latter), it cannot outcompete Na^+^ (all positive ΔGs). Adding the results evaluated for the coinage cations to those of their alkali counterparts, the order of metals’ affinity to salinomycin in low-polarity solvents becomes: Li^+^ > Cu^+^ > Na^+^ > K^+^ > Au^+^ > Ag^+^ > Rb^+^ > Cs^+^, whereas some reordering takes place in high-dielectric media: Li^+^ ≥ Na^+^ > K^+^ > Cu^+^ > Au^+^ > Ag^+^ > Rb^+^ > Cs^+^.

### 2.3. Salinomycin vs. Monensin

Recently we studied the process of complexation of monensin—another representative of ionophore antibiotics, analogous to that of salinomycin—with the same series of monovalent metal cations, where the trends of changes in metal cation affinity to the host ligand were established [35]. Although generally similar, the two ionophores exhibit some variances in their complexation behavior, differing quite significantly in their flexibility. Monensin appears to be much more rigid than salinomycin forcing the incoming metal cations (regardless of their coordination preferences) to adopt a six-coordinated geometry with octahedrally arranged oxygen-containing groups. On the other hand, in salinomycin monovalent metal complexes, it is the cation that dictates the coordination geometry. As seen (above) the coordination numbers of different metals are scattered between 2 (Au^+^), 4 (Li^+^ and Cu^+^), 5/6 (Na^+^, K^+^, Ag^+^), 6/7 (Rb^+^) and 7/8 (Cs^+^). Along the same vein, salinomycin is flexible enough allowing the metal ions to accept and incorporate as a part of their coordination sphere water molecule(s). Differences in coordination pattern reflect on the relative affinity of metal cations to the two ionophores. Although the sequence is the same for non-polar solvents (Li^+^ > Cu^+^ > Na^+^ > K^+^ > Au^+^ > Ag^+^ > Rb^+^ > Cs^+^), it is different in high dielectric media: Na^+^ > Li^+^ > K^+^ > Cu^+^ > Au^+^ > Ag^+^ > Rb^+^ > Cs^+^ (monensin) and Li^+^ ≥ Na^+^ > K^+^ > Cu^+^ > Au^+^ > Ag^+^ > Rb^+^ > Cs^+^ (salinomycin). The calculations revealed that the change in solvent polarity may favor the binding of certain metal ions and this finding can be explored under laboratory conditions. The growth of crystals suitable for X-ray diffraction studies will bring new insights into the structural chemistry of salinomycin in the solid state.

## 3. Methods

### 3.1. Gibbs Free Energy Change for the Reaction Modeled

The cation exchange free energy for Equation (1) in a medium characterized by an effective dielectric constant ε can be calculated as a sum of electronic (ΔG^1^) and solvation (ΔΔG^ε^_solv_) effects (Equation (2)):ΔG^ε^ = ΔG^1^ + ΔΔG^ε^_solv_(2)
where ΔG^1^ is the gas-phase free energy for Equation (1) (see below in next subsection), whereas ΔΔG^ε^_solv_ is the difference in the solvation free energies ΔG^ε^_solv_ of the products and reactants in the same process (Equation (3)).
ΔΔG^ε^_solv_ = ΔG^ε^_solv_ ([SalM]) + ΔG^ε^_solv_ ([Na^+^-solution]) (3)− ΔG^ε^_solv_ ([SalNa]) − ΔG^ε^_solv_ ([M^+^-solution]).

Equation (1) was modeled in different dielectric environments ranging from non-polar solvents such as cyclohexane (ε ≈ 2) and diethyl ether (ε ≈ 4) to highly polar counterparts such as methanol (ε ≈ 32) and water (ε ≈ 78). The positive Gibbs free energy change for Equation (1) suggests a Na^+^-selective ligand while the negative value implies an M^+^-selective ionophore. Note that our aim is to derive reliable trends of changes in the free energy of metal substitution (i.e., ranking the metal cations with respect to their relative affinity to salinomycin), rather than evaluating the absolute free energies of metal exchange. Such an approach has been successfully applied to various metal-containing systems such as monensin [35], macrocyclic cage molecules [37], model ion channels [38,39] and metalloproteins [40,41].

### 3.2. DFT/PCM Calculations

The Gaussian 09 suite of programs [42] was employed in performing the calculations. All the metal constructs were fully optimized in the gas phase at the B3LYP/6-31+G(d,p) level of theory producing the respective electronic energies, E_elect_. The SDD basis set and the effective core potential were used for the heavier cations from the group IA (Rb^+^ and Cs^+^) and IB (Ag^+^ and Au^+^). This combination of method/basis set was selected for the current computations as it proved reliable and performed very well in reproducing the experimental structures of a series of Na^+^/M^+^ complexes similar to salinomycin ionophore—monensin, studied in our recent work [35]. The vibrational frequencies of the metal complexes were evaluated at the same level of theory. No imaginary frequency was detected for any of the optimized constructs signifying a local minimum of the potential energy surface. The vibrational frequencies were employed in evaluating the thermal energies, including the zero-point energy, E_T_, and entropy, S. These values were used in estimating the gas phase Gibbs free energies for Equation (1), ΔG^1^, at room temperature, T = 298.15 K, according to Equation (4):ΔG^1^ = ΔE_elect_ + ΔE_T_ − TΔS(4)
where ΔE_elect_, ΔE_T_, and ΔS are the respective differences between the products and reactants. The solvation effects were assessed by employing polarizable continuum model calculations utilizing SMD scheme [43] as implemented in Gaussian 09.

Single point calculations of each gas-phase optimized structure were conducted in each solvent. The differences between the gas phase and solution energies yielded the free energies of solvation of the complex, ΔG^ε^_solv_ (Table 5 and Table 6). Notably, the experimentally determined solvation free energies for the metal cations in aqueous solution (Li^+^, Na^+^, K^+^, Rb^+^, Cs^+^, and Ag^+^) were used in the computations. For other cations (Cu^+^ and Au^+^) and solvents, the estimated values were used, where the ratios in the theoretically evaluated quantities along with the experimental values were used to determine the respective ΔG^ε^_solv_ of [M^+^/Na^+^-solution] [35]. Of note, such a calculation protocol utilizing a thermodynamic cycle that employs the experimental solvation free energies of some of participating entities (where available), has proven quite dependable in reproducing the experimental thermodynamic data [44]. The basis set superposition error (BSSE) for this type of exchange reaction (Equation (1)) was found to be negligible [38] and, therefore, was not considered in the current calculations.

## 4. Conclusions

The quantum chemical study performed disclosed the nature of the monovalent metal ions as the major factor in governing the affinity of salinomycinate anion towards group IA and IB cations. The smaller ions possessing higher positive charge density and higher ligand preference (Li^+^ and Cu^+^) predominantly coordinate in an environment with lower dielectric properties. In polar solvents, Li^+^ ions are superior to Na^+^, while the heavier alkali metal ions and their group IB counterparts are weaker competitors. The internal cavity of salinomycinate ion appears to be very flexible accommodating one or two water molecules that occupy, in addition to salinomycin binding groups, the inner coordination sphere of the metal center. The latter is crucial for the complex geometry where the metal coordination number varies from two in SalAu-W_1_/W_2_ to eight in SalCs-W_1_.

## Data Availability

Data is available from the authors upon request.

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
