# Peer review of "A DFT/PCM Study on the Affinity of Salinomycin to Bind Monovalent Metal Cations"

_molecules, 2022, doi:10.3390/molecules27020532_

Round 1

Reviewer 1 Report

In this paper, the authors investigate the affinity of the polyether ionophore salinomycin to bind IA/IB metal ions was accessed using the Gibbs free energy of the competition reaction between SalNa (taken as a reference) and its rival ions: [M+-solution] + [SalNa]  [SalM] + [Na+-solution] (M = Li, K, Rb, Cs, Cu, Ag, Au). The data is sufficient and the conclusion is reasonable. However, there is lacking in some points. Thus, in this status, I suggested its potential acceptance after a minor revise. Here are some comments as follows.

  1. Novelty of the work be established.
  2. English must be improved.
  3. In the paper, the author discussed the merit of the DFT/PCM computationsbut in the result, the discussion needs the depth… so the author should explain more about crystal structures, like single-crystal X-ray diffraction data and etc?
  4. Can the single crystalsmentioned in the article be actually synthesized based on calculations? Has it been actually verified?
  5. The advantages of crystals in practical applications.

Author Response

In this paper, the authors investigate the affinity of the polyether ionophore salinomycin to bind IA/IB metal ions was accessed using the Gibbs free energy of the competition reaction between SalNa (taken as a reference) and its rival ions: [M+-solution] + [SalNa]  [SalM] + [Na+-solution] (M = Li, K, Rb, Cs, Cu, Ag, Au). The data is sufficient and the conclusion is reasonable. However, there is lacking in some points. Thus, in this status, I suggested its potential acceptance after a minor revise. Here are some comments as follows.

Point 1. Novelty of the work be established.

Response 1. The novelty of the work was established.

Point 2. English must be improved.

Response 2. English was improved.

Point 3. In the paper, the author discussed the merit of the DFT/PCM computations but in the result, the discussion needs the depth… so the author should explain more about crystal structures, like single-crystal X-ray diffraction data and etc?

Response 3. Details on calculation protocol using the only known crystal structures of sodium salinomycin are provided in Results and Discussion Section. Salinomycin and its derivatives possess very low crystallization ability. As can be seen in CCDC Cambridge database, just 9 crystal structures of organic modifications of Salinomycin are available, and only 1 – belonging to the sodium complex of the native structure.

Point 4. Can the single crystals mentioned in the article be actually synthesized based on calculations? Has it been actually verified?

Response 4. The calculations revealed that the change of solvent polarity may favor the binding of certain metal ions and this finding can be explored at laboratory conditions.

Point 5. The advantages of crystals in practical applications.

Response 5. The growth of crystals suitable for X-ray diffraction studies will bring new insights into the structural chemistry of the antibiotic in solid state.

Reviewer 2 Report

The study represented in the manuscript is relevant, but I have some suggestions/questions for improvement.

1) How do the authors fix the positive charge on metal ions? Any charge analysis conducted?

2) As there is a possibility for dispersion, why it was neglected during the choice of level of theory and basis set?

3) Authors relied only on energy data. But I suggest using at least NBO studies along with this. This can supplement energy analysis.

4) Energy decomposition analysis will also be useful.

5) Authors claim that they have done the implicit solvent analysis using PCM. But no such studies are reported in the manuscript. No table shows the solvation data.

6) No conclusion section in the manuscript.

Author Response

The study represented in the manuscript is relevant, but I have some suggestions/questions for improvement.

Point 1. How do the authors fix the positive charge on metal ions? Any charge analysis conducted?

Response 1. The Gaussian program package as well as other similar programs for quantum-chemical calculations do not fix the individual atomic charges a priori. As an input, geometry, the overall charge and electron multiplicity of the molecule/complex are given.

Point 2. As there is a possibility for dispersion, why it was neglected during the choice of level of theory and basis set?

Response 2. Atomic charges, after structure optimization, are evaluated from the electron density distribution in the system by employing different schemes such as NBO, Hirshfeld and CM5. Furthermore, interactions between the metal cation and host salinomycin are electrostatic in nature and dominate the energetics of the system over the much weaker dispersion interactions. Therefore, we believe that employing B3LYP functional, which has proven reliable in evaluating properties of many metal containing systems, is appropriate for solving the problems envisaged in the present study.

Point 3. Authors relied only on energy data. But I suggest using at least NBO studies along with this. This can supplement energy analysis.

Response 3. We have added tables containing data from NBO, Hirshfeld and CM5 population analyses (Tables 3-4) which corroborate very well the Gibbs free energies of M+ / Na+ metal substitution.

Point 4. Energy decomposition analysis will also be useful.

Response 4. Gaussian suite of programs that we use does not have an option to perform such an analysis. Moreover, other packages (GAMESS for example), as far as we know, offer such a possibility for HF but not for DFT calculations employed here. These factors precluded performing decomposition analysis in the present paper.

Point 5. Authors claim that they have done the implicit solvent analysis using PCM. But no such studies are reported in the manuscript. No table shows the solvation data.

Response 5. We have added data from PCM calculations (Tables 5-6).

Point 6. No conclusion section in the manuscript.

Response 6. Conclusion section is now included.

Round 2

Reviewer 2 Report

The authors explanations are adequate and they have improved the original manuscript.

Hence ACCEPT the manuscript for publication.